# The Effect of a Combined Gluten- and Casein-Free Diet on Children and Adolescents with Autism Spectrum Disorders: A Systematic Review and Meta-Analysis

**DOI:** 10.3390/nu13020470

**Published:** 2021-01-30

**Authors:** Amélie Keller, Marie Louise Rimestad, Jeanett Friis Rohde, Birgitte Holm Petersen, Christoffer Bruun Korfitsen, Simon Tarp, Marlene Briciet Lauritsen, Mina Nicole Händel

**Affiliations:** 1Section of Epidemiology, Department of Public Health, University of Copenhagen, 1014 Copenhagen, Denmark; 2Institute of Psychology, Aarhus University, 8000 Aarhus, Denmark; psykolog-rimestad@protonmail.com; 3The Parker Institute, Bispebjerg and Frederiksberg Hospital, The Capital Region, 2000 Frederiksberg, Denmark; jeanett.friis.rohde@regionh.dk (J.F.R.); CBKO@SST.DK (C.B.K.); sita@SST.DK (S.T.); Mina.Nicole.Holmgaard.Handel@regionh.dk (M.N.H.); 4The Danish Health Authority, 2300 Copenhagen, Denmark; bhp@sst.dk; 5Psychiatry, Aalborg University Hospital and Department of Clinical Medicine, Aalborg University, 9000 Aalborg, Denmark; marlene.lauritsen@rn.dk

**Keywords:** autism spectrum disorder, gluten-free/casein-free diet, childhood

## Abstract

There has been a growing interest in the gastrointestinal system and its significance for autism spectrum disorder (ASD), including the significance of adopting a gluten-free and casein-free (GFCF) diet. The objective was to investigate beneficial and safety of a GFCF diet among children with a diagnosis of ASD. We performed a systematic literature search in Medline, Embase, Cinahl, and the Cochrane Library up to January 2020 for existing systematic reviews and individual randomized controlled trials (RCTs). Studies were included if they investigated a GFCF diet compared to a regular diet in children aged 3 to 17 years diagnosed with ASD, with or without comorbidities. The quality of the identified existing reviews was assessed using A Measurement Tool to Assess Systematic Reviews (AMSTAR). The risk of bias in RCTs was assessed using the Cochrane Risk of Bias Tool, and overall quality of evidence was evaluated using Grades of Recommendation, Assessment, Development, and Evaluation (GRADE). We identified six relevant RCTs, which included 143 participants. The results from a random effect model showed no effect of a GFCF diet on clinician-reported autism core symptoms (standardized mean difference (SMD) −0.31 (95% Cl. −0.89, 0.27)), parent-reported functional level (mean difference (MD) 0.61 (95% Cl −5.92, 7.14)) or behavioral difficulties (MD 0.80 (95% Cl −6.56, 10.16)). On the contrary, a GFCF diet might trigger gastrointestinal adverse effects (relative risk (RR) 2.33 (95% Cl 0.69, 7.90)). The quality of evidence ranged from low to very low due to serious risk of bias, serious risk of inconsistency, and serious risk of imprecision. Clinical implications of the present findings may be careful consideration of introducing a GFCF diet to children with ASD. However, the limitations of the current literature hinder the possibility of drawing any solid conclusion, and more high-quality RCTs are needed. The protocol is registered at the Danish Health Authority website.

## 1. Introduction

Autism spectrum disorder (ASD) is a neurodevelopmental disorder characterized by qualitatively impaired behavior within the core areas of social interaction and communication as well as behavior characterized by a pattern of limited, stereotypical, and repetitive actions and interests [1,2]. The impairments are persistent and pervasive and broadly affect the functioning of the individual in all contexts of life. The symptoms are present early in a child’s development but may in some cases be more distinct with time, sometimes as demands on the child and adolescent increases or when the learned compensatory strategies are no longer sufficient. ASD is believed to be a lifelong disability, but the clinical presentation and level of functioning may change over time [1,3]. 

As of now, there is no curative treatment for ASD, but in the last couple of decades, the role of the gastrointestinal system in the development of ASD has been a topic of interest, based on the finding of the high prevalence of gastrointestinal problems and disorders in individuals with ASD [4,5]. Furthermore, children with ASD have been found to exhibit higher levels of proinflammatory cytokines following exposure to food proteins from gluten, casein, and soy, compared with controls [6], findings that have stimulated research investigating the link between food allergy and ASD [7,8]. Theories of mechanisms of action of the involvement of the immune system and the gastrointestinal system in the development of ASD are many and include the involvement of the gut-blood-brain barrier where by-products of the gut microbiota like lipopolysaccharides and short-chain fatty acids have been proposed to modulate the cytokine production [9]. Also, by-products from the microbiota have been suggested to influence the synthesis of neuropeptides like serotonin, and peptides from gluten and casein have been hypothesized to increase the activity of the opioid system [10]. These neuropeptides are thought to result in impairments in social behavior and communication and thus may be involved in the pathogenesis of ASD. 

Besides being hypothesized to be involved in the development of ASD [4,11], gluten- and casein-free diet started to be used by some families a couple of decades ago as a treatment for symptoms of ASD as well as for gastrointestinal symptoms based on case reports [11,12,13,14,15,16,17,18]. However, a Cochrane review from 2008 [19], based on two randomized controlled trials [20,21], concluded that the evidence for the efficacy of eliminating gluten and casein from the diet, purely based on autism as an indication, is uncertain. As research is still needed to elucidate the pathophysiologic mechanisms behind the relationship between ASD and gluten and casein diet, recommendations on diet restrictions are presently limited to individuals where allergic reactions or intolerance have been detected irrespective of a diagnosis of autism [22]. Meanwhile, research has shown that the use of complementary and alternative therapies for children and adolescents, including gluten-free and casein-free (GFCF) diet, are widely used in real-life settings [19], even though the treatment strategy is resource-demanding for the child with ASD and their family. Thus, to provide clinicians and guideline panels as well as caregivers with an updated overview of the current evidence from randomized controlled trials, the objective of this systematic review and meta-analysis was to synthesize the literature on the effectiveness and side effects of a GFCF diet compared to regular diet among children aged 3 to 17 years with a diagnosis of ASD, with or without comorbidities in randomized trials, and to rate the certainty of the current evidence. 

## 2. Methods

This systematic review and meta-analysis was performed according to the principles described in the Grades of Recommendation, Assessment, Development, and Evaluation (GRADE) approach [23], and adheres to the Preferred Reporting Items for Systematic Review and Meta-analysis (PRISMA) [24,25,26] (PRISMA checklist is provided in the Appendix A), as well as the guidelines of the Cochrane Collaboration [27]. Moreover, this systematic review was structured according to Population, Intervention, Comparison, and Outcome (PICO) characterization [28]. The work serves as a part of national clinical guidelines on the treatment of ASD among children (<18 years) published by the Danish Health Authority in 2021. As such, the study protocol (in Danish) was pre-specified, registered, and approved by the management at the Danish Health Authority in November 2019 (available at the Danish Health Authority website: www.sst.dk).

### 2.1. Search Strategy 

The search was conducted throughout January 2020 (last date of search was 24 January 2020) in two steps: (1) systematic reviews, meta-analysis with a filter, (2) primary literature—randomized controlled trials (RCTs) with a filter. The reason for the two-step approach is to accelerate the search process and thus identify primary literature from existing high-quality systematic reviews. The databases searched were Medline, Embase, Cinahl, and Cochrane Library via Medline and Embase. The keywords for the search were applied as controlled vocabulary and free text. The full electronic search strategy for all databases, including the limits used and dates of coverage, are presented in detail in Appendix A: Search description. Moreover, content experts from the guideline working group were conferred whether any studies were missing from the search, and reference lists of included articles and previous reviews were hand-searched for potentially relevant studies. Study authors were not contacted to identify additional studies.

### 2.2. Study Selection 

The studies generated from the defined search strategy on both systematic reviews and individual RCTs were imported to RefWorks, where duplicates were removed. Hereafter the remaining references were imported into Covidence software for literature screening and data management. One reviewer (M.L.R) evaluated the titles and abstracts for the articles following the pre-specified criteria. Full texts identified in the first step were screened independently by two reviewers (M.L.R. and M.N.H.). Disagreements were resolved through discussion. Reference lists of selected articles were also searched. 

To be included in the systematic review and meta-analyses, the included studies had to fulfill the following eligibility criteria:

#### 2.2.1. Population

Children of 3–17 years of age diagnosed with ASD according to the diagnostic criteria of the Diagnostic Statistical Manual (DSM) or the International Classification of Diseases (ICD) with or without comorbidities. This age range was chosen to cover children and adolescents with ASD from the earliest age of a valid diagnosis (3 years of age) until the eighteenth birthday, when adolescents in Denmark come of age.

#### 2.2.2. Intervention

Diet with the elimination of both gluten and casein. In order to be able to observe a potential change in autistic symptoms, a sufficient period is needed for the symptoms to diminish in severity. Regarding the length of follow-up, a minimum duration of six months was, from a clinical point of view, considered enough for this to occur.

#### 2.2.3. Comparator

No treatment—regular diet.

#### 2.2.4. Outcomes


***Primary outcome***


Core autistic symptoms, clinician-/observer-reported.


*Timing and effect measures*


The primary outcome was investigated at the end of treatment (minimum six months from baseline).

In crossover trials, to limit the potential presence of carryover effects of treatment from the first period, the initial period before crossover was used [27].


***Secondary outcomes***


Adaptive functional level in the child, clinician-reported.Adaptive functional level in the child, parent-reported.Core autistic symptoms, parent-reported.Gastrointestinal discomfort.Behavioral difficulties, clinician-/observer-reported.Behavioral difficulties, parent-reported.Other side effects, such as constipation, irregular bowel, diarrhea, stomach aches, sleep disturbances, and appetite disturbances.Weight change.Child quality of life, parent-reported.Parental well-being.


*Timing and effect measures*


The secondary outcomes were investigated at the end of treatment (minimum six months from baseline).

### 2.3. Study Design

RCTs, including crossover design, were considered for this review.

### 2.4. Report Characteristics

All years were considered, and there was no restriction on publication status, i.e., conference abstract was considered, if the results were not already published. Only RCTs reported in English or in a Scandinavian language were considered. 

### 2.5. Data Extraction of Individual Randomized Trials 

Data extraction was conducted independently and in duplicate by two out of three reviewers (C.B.K., J.F.R., and M.N.H.) using a predefined template in Covidence software [29]. Additionally, the following information on the descriptive and quantitative characteristics of studies were extracted: (i) characteristics of the study: authorship, year, country, setting, sample size, design, methods, duration of follow-up, source of funding, and conflict of interest; (ii) characteristics of the population: age, ethnicity, co-interventions, information regarding respondent bias, or representativeness of included population; (iii) details about the intervention; (iv) details about comparator group (e.g., placebo); (v) outcomes: as above-mentioned.

Study authors were not contacted in relation to confirming data, no assumptions and simplifications were placed on data, and thus, study information on data items was used as presented by study authors. 

### 2.6. Quality Assessment

At the study level, the quality of included systematic reviews and individual studies was assessed independently and in duplicate by two out of three reviewers (C.B.K., J.F.R., and M.N.H.) based on (a) A Measurement Tool to Assess Systematic Reviews (AMSTAR) [30] and (b) the criterion provided by the Cochrane Collaboration’s tool for assessing the risk of bias of RCTs [31]. The AMSTAR tool provides 11 quality domains, and each domain is classified into four levels of risk of bias (yes, no, not clear, or not applicable), with yes indicating a low risk of bias. The Cochrane Collaboration’s tool provides seven quality domains [20]. Each domain is classified into three levels of risk of bias (low, high, or unclear) based on specific criteria. The seven domains are as follows: sequence generation, allocation concealment, blinding of participants and personnel, blinding of outcome assessment, incomplete outcome data, selective outcome reporting, and other sources of bias. 

At the outcome level, we assessed the certainty in the evidence of each outcome using the GRADE approach [23], which categorized each estimate as very low, low, moderate, or high and is an indication of the robustness in the interpretations of the results. RCTs started at a high certainty level and were then assessed for possible downgrading based on the following domains: overall risk of bias, inconsistency, indirectness, imprecision, and publication bias.

The above-mentioned quality assessments were used in the data synthesis to address the methodological limitations and the influence it may have on the results in terms of certainty of the evidence. 

### 2.7. Meta-Analysis

For all outcomes that were reported as a continuous variable, the effect size was assessed as the mean difference (MD) (95% confidence interval (CI)) if the presented data were reported using the same measurement scale. If different measurement scales were used, the effect size was calculated using a standardized mean difference (SMD; 95% CI). Dichotomous outcomes were analyzed by calculating the relative risk (RR; 95% CI). To include studies that had zero events in both intervention and control group, a risk difference (RD; 95% CI) was calculated as a sensitivity analysis. A random-effect model was applied for all the meta-analyses. Statistical heterogeneity was quantified using I^2^ statistics [32].

Review Manager Software (version 5.3) (The Nordic Cochrane Centre, The Cochrane Collaboration, Copenhagen, Denmark) was used to produce the analyses and forest plots.

A priori, we planned to assess publication bias through funnel plots, but since only a few studies were included, the power of the tests is too low to distinguish chance from real asymmetry, and therefore these analyses were not performed [27]. 

## 3. Results

### 3.1. Literature Search 

In the search for systematic reviews, 654 references were identified. Following the removal of duplicates and non-relevant references, we screened 20 records at full-text level, and finally, we identified three systematic reviews [33,34,35] (Figure 1).

The AMSTAR quality assessment of the three systematic reviews can be found as Appendix A. References of the three included systematic reviews contributed with five RCTs [20,21,36,37,38] (Figure 2).

A follow-up search for supplementary primary literature was conducted from January 2016 (based on the search date from Piwowarzcyk et al. (2018) [35]) to January 2020, where we identified one more RCT [39]. Thus, the total body of evidence in this review is based on six RCTs [20,21,36,37,38,39] (Figure 3).

List of excluded studies after full-text screening, including reasons for their exclusion, is presented in the Appendix A.

### 3.2. Description of the Primary Studies

The six studies [20,21,36,37,38,39] that met the criteria to be included in the present review were all RCTs conducted between 2006 and 2019. The characteristics of the six RCTs are summarized in Table 1, Table 2 and Table 3. Three RCTs were conducted in the United States of America (USA) [21,36,38] and the others in Spain [39], Norway [20], and Denmark [37], respectively (Table 1). A total of 178 children aged between 2 and 9 years participated in the included studies. Due to few numbers of primary studies, we decided to include studies with mean ages below 3 years (Table 2). Two RCTs were double-blinded (blinding of participants (subjects and parents), personnel and of outcome assessment [21,38], three RCTs were single-blinded (blinding of outcome assessment [36], blinding of participants [20], blinding of outcome assessment [37] and no blinding was applied in the study by Gonzalez-Domenech et al. (2019) [39] (Table 1). Two were crossover trials (where the initial period before crossover was used) [21,39], and the remaining four RCTs included parallel groups [20,36,37,38] (Table 1).

The interventions consisted of the elimination of gluten and casein from the diet, which varied in duration, from four weeks to one year. In four studies [20,36,37,39], parents received instructions on how to provide a GFCF diet to their child. In the study by Navarro et al. (2015) [38] participants followed two weeks of a GFCF diet followed by four weeks of a GFCF diet + supplement containing brown rice flour, whereas in the study by Elder et al. (2006) [21] a GFCF diet was provided bi-weekly (Table 3). 

The comparison groups received a regular diet in four studies [20,21,37,39], a diet without sugar in one study [36], and a GFCF diet with a dietary supplement containing 0.5 g/kg of gluten powder and 0.5 g/kg of non-fat dried milk in another RCT [38] (Table 3).

### 3.3. Synthesis of Results of Primary Studies

#### 3.3.1. Primary Outcome: Clinician-Assessed Core Symptoms

Four studies [20,21,37,39] assessed the association between GFCF diet and clinician-reported core symptoms of ASD, and a statistically non-significant effect was shown (SMD of −0.31 (95% Cl. −0.89, 0.27)) (Figure 4). There was moderate heterogeneity (I^2^ = 54%). 

#### 3.3.2. Secondary Outcomes

The associations between GFCF diet and adaptive functioning in the child (parent-reported), behavioral difficulties (parent-reported) [36], weight [39], and gastrointestinal discomfort [25,37] were each assessed by one study, respectively. Two RCTs reported on all side effects [36,37]. Some increased incidence in the number of people with other side effects (RR 1.89 (95% Cl. 1.11, 3.21) and RD 0.23 (95% Cl. −0.26, 0.72)) (Figure 5A,B) with high degree of heterogeneity (I^2^ = 91%), and gastrointestinal discomfort (RR 2.33 (95% Cl. 0.69, 7.90)) (Figure 6) was reported in the GFCF treatment group. Other reported side effects besides gastrointestinal discomfort and weight loss were waking up at night and decreased appetite. No clinically relevant effect was seen on either parent-reported functional level of the child (MD 0.61 (95% Cl. −5.92, 7.14)) or child’s behavioral difficulties (MD 0.80 (95% Cl. −6.56, 10.16)) (Figure 7 and Figure 8). 

For weight change, measured as body mass index (BMI, kg/m^2^), there was no clinically relevant effect (MD 0.30 (95% Cl. −1.81, 2.41)) (Figure 9).

No studies described parental-reported autism core symptoms, clinician-assessed adaptive functioning of the child, clinician-reported child’s behavioral difficulties, quality of life of the child, or parental well-being. 

### 3.4. Certainty of Evidence (GRADE)

In summary, for both the primary and secondary outcomes, the quality of evidence ranged from low to very low due to serious risk of bias, serious risk of inconsistency, and serious risk of imprecision. A summary of findings can be found in Table 4.

The Cochrane risk-of-bias domains which presented the highest risk of bias were (1) blinding of participants and personnel (performance bias), (2) incomplete outcome data (attrition bias), and (3) selective reporting (reporting bias), where half or more than half of the RCTs presented a high risk of bias. The risk of selection bias, assessed by sequence generation and allocation concealment, was unclear in four out of the six included RCTs (Figure 10). 

The quality of the evidence for the primary outcome clinician-assessed core symptoms was very low as two out of four studies included in this analysis had a high risk of bias due to lack of blindness of participants and evaluators of the effect, as well as incomplete data due to a large dropout, severe inconsistent results due to high heterogeneity and severe non-transferability due to insufficient time frame. In addition, the few children participating (*n* = 120) in the analysis resulted in a severely inaccurate effect estimate. 

For the secondary outcomes other side effects and gastrointestinal discomfort, the confidence in the estimates was very low due to a lack of blindness of participants and evaluators, as well as incomplete data due to a large dropout rate, severe non-transferability due to insufficient time frame, inaccurate effect estimate due to few children participating and high heterogeneity resulting in high risk of bias. Thus, it is uncertain whether GFCF diets increase the incidence of side effects due to the very low confidence in the estimates. For the parent-reported functional level, there was a high risk of bias due to a lack of blindness of participants and assessors and possible selective reporting of outcomes, as well as inaccurate effect estimate with few (*n* = 55) participants from a single study [37]. For parent-reported behavioral difficulties, downgrades had to be made for the same parameters as for the above, as well as for the risk of attrition bias. Very low confidence in the estimate due to serious risk of bias for the secondary outcome weight change was also present, as selection, detection, and attrition bias were high, and the effect estimate was based on few participants (*n* = 20) from a single study [39].

## 4. Discussion

The objective of this systematic literature review and meta-analysis was to provide clinicians and guideline panels as well as caregivers with a special interest in this field with an updated overview and to critically assess the evidence investigating the effect of a GFCF diet among children aged 3 to 17 years with a diagnosis of ASD. Based on the collective evidence from the six identified RCTs [20,21,36,37,38,39], there are no indications that following a GFCF diet has a positive effect on core symptoms of autism, behavioral difficulties, or adaptive functional level, and there is a high degree of uncertainty due to very low quality of evidence. 

During the search, three systematic reviews [33,34,35] that investigated the effect of GFCF diet among children with ASD were identified, which included five [35], four [22,33], and three [34] of the six RCTs [20,21,36,37,38,39] included in the present review. Previous reviews highlighted the fact that uncontrolled trials or case reports tend to show a significant improvement of ASD symptoms after the elimination of gluten and/or casein from the diet. However, these studies suffered from methodological flaws such as small sample sizes, absence of control groups, use of unstandardized outcome measures, lack of blinding, and poor ASD diagnosis characterization [33,34,35]. 

The findings of reviews based on RCTs on this subject are consistent with our findings. Previous evidence based on results from RCTs has shown that adherence to a GFCF diet was not associated with an improvement in ASD outcomes [33,34,35]. With respect to the primary outcome measure defined in this review, i.e., clinician-assessed core autistic symptoms, it is noteworthy that this outcome measure was also investigated in all but one of the six RCTs. However, the methods used to measure core symptoms of autism differ between studies, and the widely used diagnostic instrument in research and clinical practice, The Autism Diagnostic Observation Schedule (ADOS) [40,41], was only used in two of the RCTs [36,37]. However, blinding regarding the primary outcome measure has been performed in most studies. 

The present review suggests that a GFCF diet may cause some increased incidence of gastrointestinal adverse effects, which is important to bear in mind when choosing to put a child on a gluten- and/or casein-free diet, which is often performed by parents in the hope of reducing or even eliminating autistic symptoms. Also, it is well-known that individuals with ASD commonly present selective eating patterns that may worsen when introducing a diet with subsequent risk of eating disorder and/or malnutrition [42,43,44]. 

Besides gastrointestinal side effects, it is worth noticing the risk of decreased appetite and weight loss in addition to waking up at night when introducing a gluten- and casein-free diet. Sleep disturbances are common in children with ASD [22], and the addition of diet restrictions to a child with ASD may worsen sleep difficulties and subsequently worsen the well-being of the child. The finding of a risk of weight loss is important information for clinical practice, and the use of anthropometric growth measures (weight and height and calculation of BMI) when evaluating the effect of diet restriction should be emphasized.

The lack of efficacy of the diet may be explained by low adherence to the diet restriction due to severity of ASD in the child and/or parental difficulties with supplying GFCF food, possibly due to the added burden of a strict elimination diet in addition to the often stressed family life of having children with ASD [45,46]. Five out of six included studies monitored adherence to diet [21,36,37,38,39], mainly relying on parental reports, which are highly subjective to selective reporting. In two included studies [21,39], where urinary concentrations of peptide concentrations were monitored in addition to parental monitoring of adherence to the diet, there was no significant decrease in peptide concentrations following a GFCF diet, suggesting either contamination or no efficacy. This warrants caution regarding the conclusions that can be drawn from the null findings. It may be noteworthy that if strict adherence is hard for children and families to follow, causing no effect of the elimination diet, this could point to a potential problem of the feasibility of the diet itself. 

Clinical implications of the present findings may be careful consideration of introducing a GFCF diet to families with children with ASD, unless intolerance or allergy towards gluten and/or casein has been suspected or detected, based on standardized and validated measures. 

Clinicians may inform families of the paucity of evidence of beneficial effects, and the increased incidence of gastrointestinal adverse effects should be taken into consideration, as well as the overall strain on the family of the added burden of introducing an elimination diet, which may be difficult to adhere to. 

### Strengths and Limitations

This systematic review and meta-analyses were performed using transparent methods and a priori defined criteria following the guidelines of the Cochrane Collaboration and PRISMA, including protocol registration at the Danish Health Authority (in Danish), comprehensive search, independent and duplicate full-text study selection, data extraction, and quality assessment. Limitations included an exclusive selection of studies published in English and Scandinavian languages and the inclusion of RCTs, which were of generally low quality. The authors of the included studies were not contacted for further information, and the grey literature was not searched; thus, the results are solely based on data published in peer-reviewed articles. 

The duration of the intervention was below the pre-specified sixmonth period in half of the included studies [21,36,38], and one [20] of the two [20,37] studies with the longest follow-up (12 months) reported a significant effect of following a GFCF diet on core autistic symptoms. Therefore, the absence of effect reported in this systematic review and meta-analysis might be owed to short intervention durations in the included studies. In addition, all secondary outcomes were addressed by one or two studies only; therefore, caution is required when drawing conclusions. Future research should focus on conducting large-scale clinical trials of high-quality, following the CONSORT (Consolidated Standards of Reporting Trials) Statement, with an adequate duration time of the GFCF diet (>6 months). Researchers should carefully monitor participant adherence to the diet. 

## 5. Conclusions

Based on the current evidence, there seems to be no benefit of providing a GFCF diet to children and adolescents with ASD concerning clinician-reported autism core symptoms or parent-reported functional level and behavioral difficulties. On the contrary, a GFCF diet might trigger gastrointestinal adverse effects. These results are consistent with the conclusions from previous reviews. The limitations of the current literature hinder the possibility of drawing any solid conclusion, and more well-designed, high-quality clinical trials of sufficient duration are required.

## Figures and Tables

**Figure 1 nutrients-13-00470-f001:**
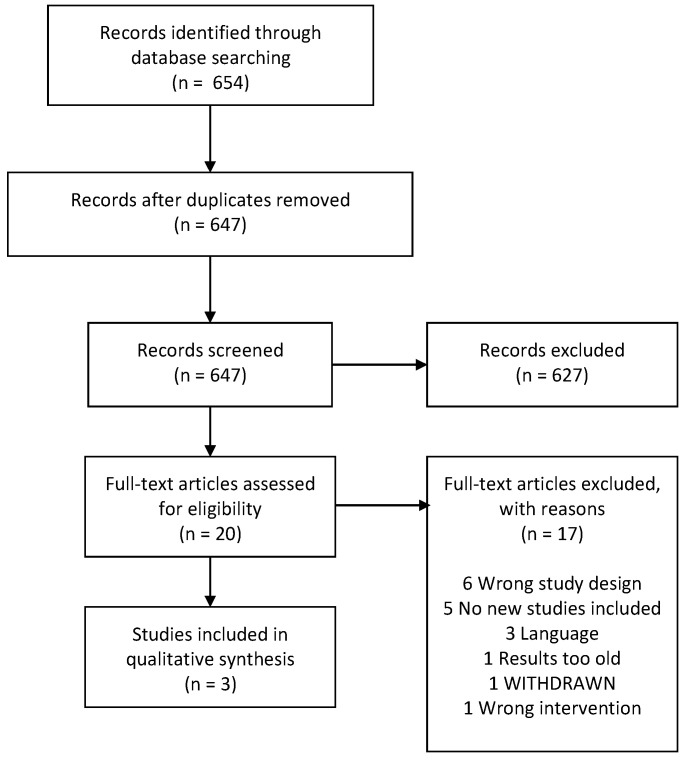
Flowchart of the systematic reviews.

**Figure 2 nutrients-13-00470-f002:**
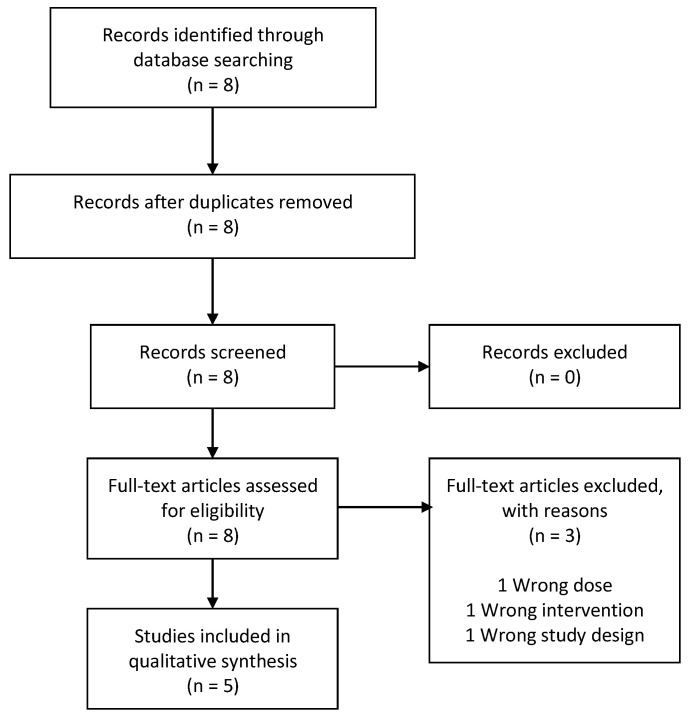
Flowchart of secondary studies identified in reviews.

**Figure 3 nutrients-13-00470-f003:**
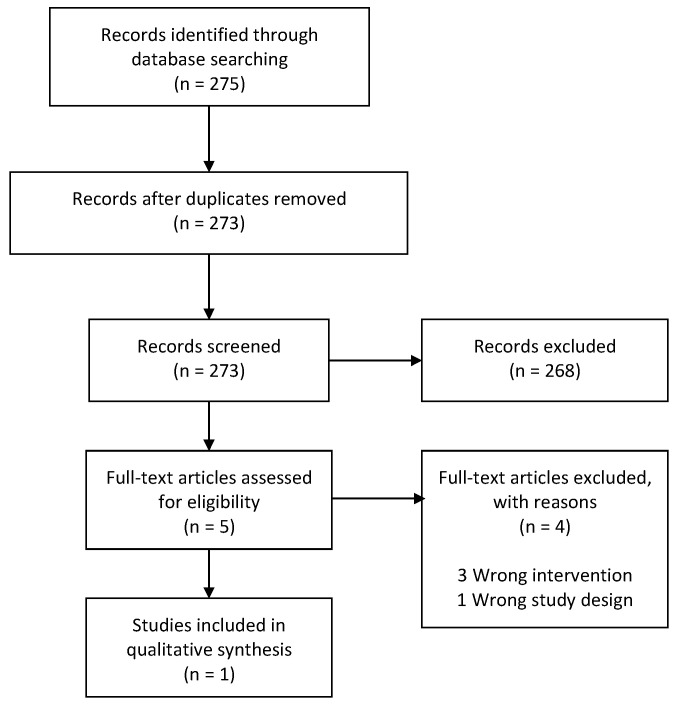
Flowchart of primary studies.

**Figure 4 nutrients-13-00470-f004:**
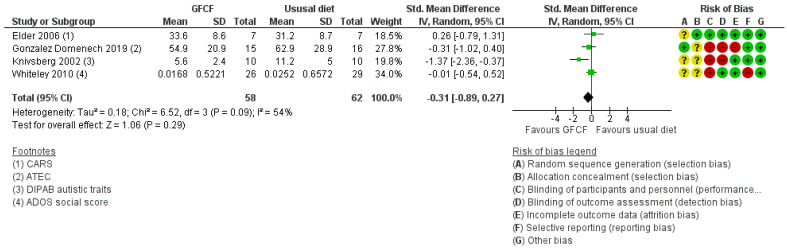
Forest plot of comparison: GFCF vs. usual diet, outcome: clinician-reported core symptoms.

**Figure 5 nutrients-13-00470-f005:**
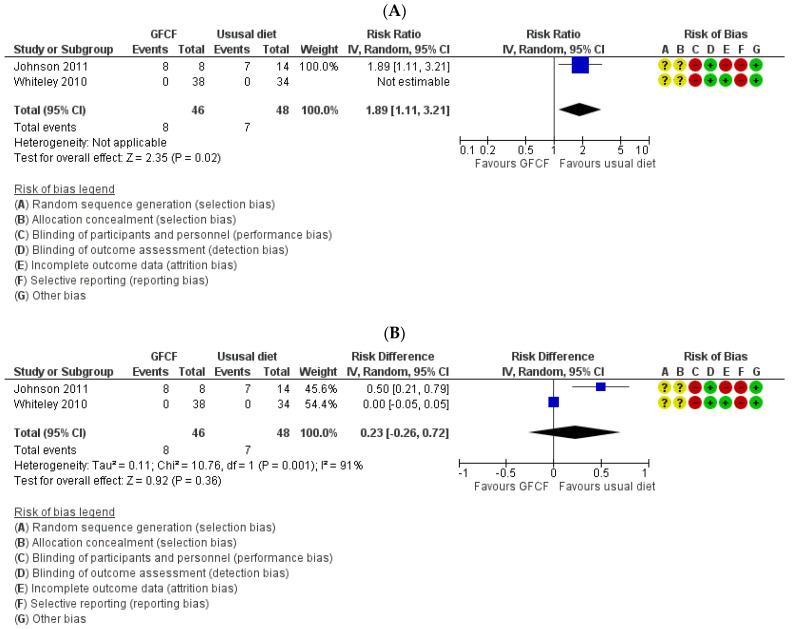
Forest plot of comparison: GFCF vs. usual diet, outcome: number of children with adverse events/side effects. (**A**) Risk ratio; (**B**) Risk difference.

**Figure 6 nutrients-13-00470-f006:**
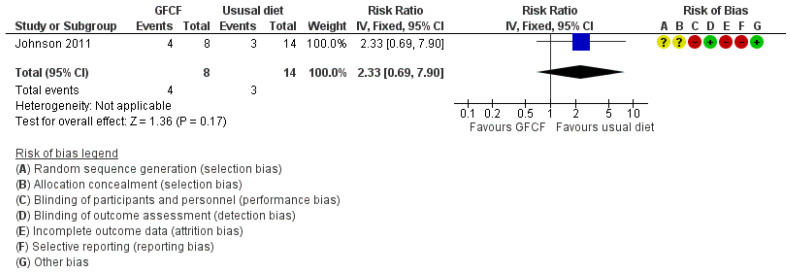
Forest plot of comparison: GFCF vs. usual diet, outcome: number of children with gastrointestinal discomfort.

**Figure 7 nutrients-13-00470-f007:**
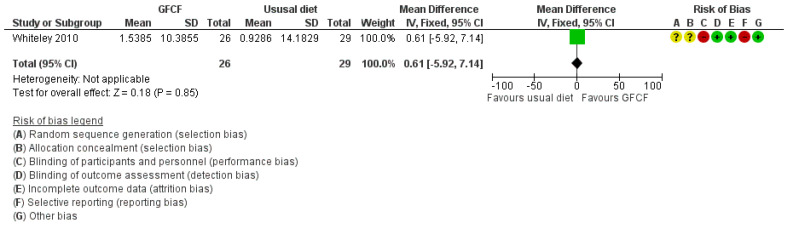
Forest plot of comparison: GFCF vs. usual diet, outcome: parent-reported functional level (VABS daily living, higher is better).

**Figure 8 nutrients-13-00470-f008:**
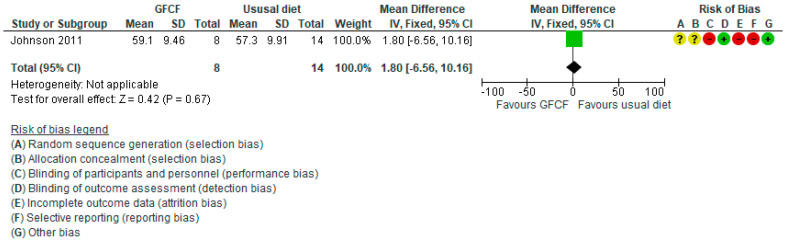
Forest plot of comparison: GFCF vs. usual diet, outcome: parent-reported child’s behavioral difficulties (CBCL Aggression subscale, lower is better).

**Figure 9 nutrients-13-00470-f009:**
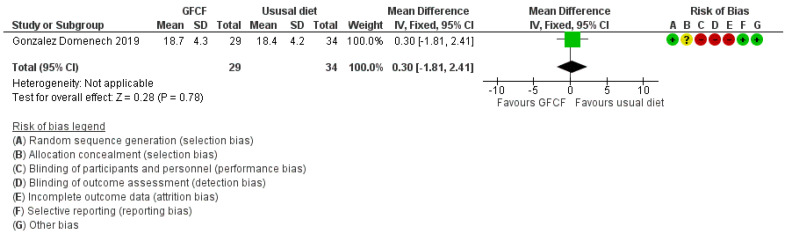
Forest plot of comparison: GFCF vs. usual diet, outcome: body mass index.

**Figure 10 nutrients-13-00470-f010:**
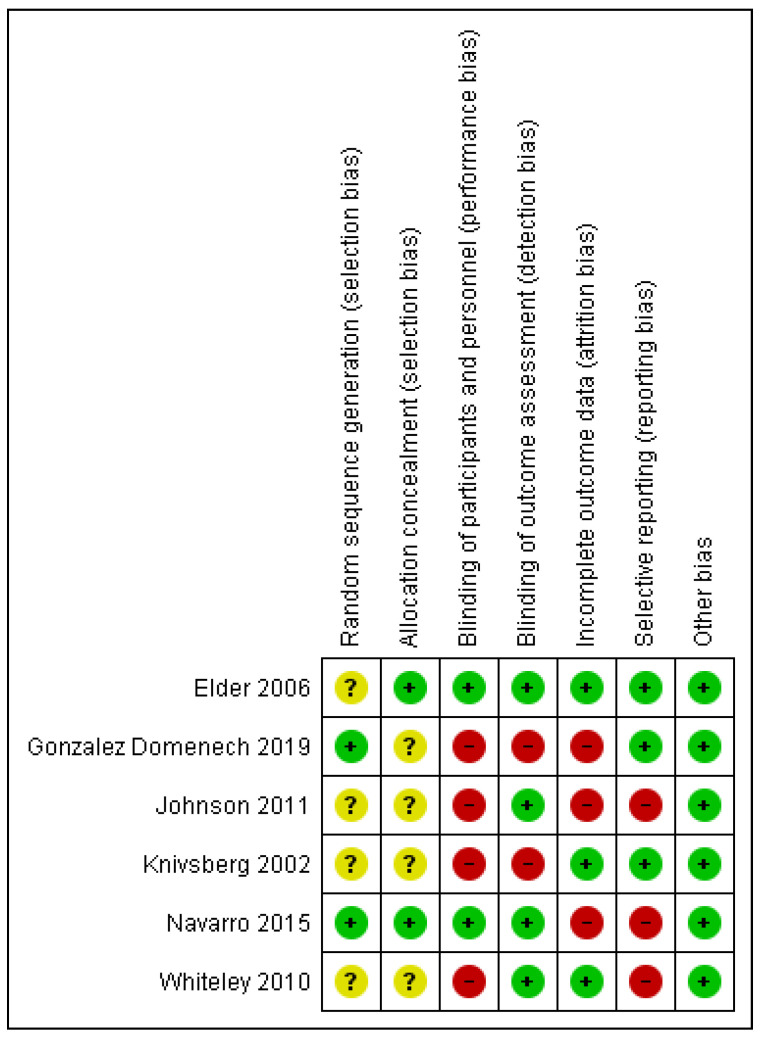
Risk of bias assessment as assessed by the Cochrane risk of bias tool. A plus (+) indicates a low risk of bias, a question mark (?) indicates an unclear risk of bias, and a minus (−) indicates a high risk of bias. The specific type of bias is presented in the top column and the individual studies in the left row.

**Table 1 nutrients-13-00470-t001:** Study identification of the six included RCTs.

Study’s First Author, Year	Region, Country	Trial Registration	Study Design	Conflict of Interest/Sponsorship
Elder et al., 2006 [21]	Florida, USA	Not reported	Double-blinded, crossover RCT	Conflict of interest: Not reported University of Florida s College of Nursing Biobehavioral NINR-funded research grant P20 NR 07791-03 and GCRC grant M01RR00082 from the National Institute of Research Resources, National Institutes of Health
Gonzalez-Domenech et al., 2019 [39]	Jaen, Granada, Malaga and Almeria, Spain	Not reported	Crossover RCT	The authors declare that they have no conflict of interest.
Johnson et al., 2011 [36]	Pennsylvania, USA	Not reported	Single-blinded RCT with parallel group	John F. and Nancy A. Emmerling Fund/The Pittsburgh Foundation. Financial Disclosures Dr. Benjamin Handen disclosed consulting fees for Forest, Bristol Myer Squibband has research funding from Forest, Bristol Myer Squibb, Johnson and Johnson, Neuropharm, and Curemark.
Knivsberg et al., 2002 [20]	Stavanger, Norway	Not reported	Single-blinded RCT with parallel group	Not reported
Navarro et al., 2015 [38]	Texas, USA	Not reported	Double-blinded RCT with parallel group	Not reported
Whiteley et al., 2010 [37]	Copenhagen, Denmark	NCT00614198 Registered in 2008	Single-blinded RCT with parallel group	Conflict of interest reported. This study was supported by the Center for Autisme, the Nils O. Seim Family Fund for Medical Research, the Eric Birger Christensen Fond, the Norwegian Protein Intolerance Association, and the Robert Luff Foundation.

Abbreviations: RCT: Randomized Controllled Trial.

**Table 2 nutrients-13-00470-t002:** The baseline characteristic of the six included RCTs.

Study’s First Author, Year	N (% Female)	Age (Mean ± SD) (Median (IQR))	Inclusion Criteria	Exclusion Criteria
Elder et al., 2006 [21]	Intervention: 7 Control: 7 (20% Female)	7.3 ± 4.1 years	Diagnosis of ASD according to DSM-IV criteria	Children with medical histories and/or physical examinations indicated that they had physical or sensory-impairments or significant medical problems, including celiac disease.
Gonzalez-Domenech et al., 2019 [39]	Intervention: 15 Control: 16 (46% Female)	Overall 8.9 years	Diagnosis of ASD according to the tenth edition of the International Classification of Diseases (ICD-10).	Patients diagnosed with an allergy to gluten or casein; patients who had previously excluded gluten and/or casein from their diet; patients who were likely to not adhere to the diet properly.
Johnson et al., 2011 [36]	Intervention: 8 Control 14 (22% Female)	Intervention 40.1 ± 9.3 months Control 39.5 ± 8.7 months	Diagnosis of ASD according to DSM-IV criteria	Not reported
Knivsberg et al., 2002 [20]	Intervention: 10 Control: 10 (Not reported)	Intervention 91 months Control 86 months	Participation included only children with both the diagnosis of autism and abnormal urinary peptide patterns.	Not reported
Navarro et al., 2015 [38]	Intervention: 6 Control: 6 (Not reported)	Intervention 5.5 years Control 6 years	Diagnosis of ASD according to DSM-IV criteria	Children with food allergies, celiac disease, inflammatory bowel disease, and infectious GI diseases. Children with neurological problems that could interfere with proper evaluation of behavior. Children with parents unwilling to undertake the GDF diet during the study.
Whiteley et al., 2010 [37]	Intervention: 38 Control: 34 (11% Female)	Intervention 94.2 (IQR 76.6–118) months Control 69.4 (ICR 76.3–120.3) months	Children diagnosed with pervasive developmental disorder (ICD-10 code F84)	Children with co-morbid diagnoses of epilepsy, fragile-X syndrome, tuberous sclerosis, or a developmental age below 24 months

Abbreviations: ASD: Autistic Spectrum Disorder; DSM-IV: Diagnostic and Statistical Manual ICD: International Classification of Diseases; GI: Gastrointestinal.

**Table 3 nutrients-13-00470-t003:** The intervention, control, reported outcomes and authors conclusion of the six included RCTs.

Study’s First Author, Year	Intervention (Duration)	Monitoring Tools	Control Group	Reported Outcomes	Authors Conclusion
Elder et al., 2006 [21]	GFCF experimental diet (6 weeks)	Parents received a 3- to 4-day supply of food twice each week + a list of allowed foods in case of emergencies and they were asked to record their child’s diet intake to monitor compliance. Pre and post urinary concentration of casein , gluten peptides, casomorphin and gliadorphin	Regular diet.	Core autistic symptoms, clinician-reported	Null findings
Gonzalez-Domenech et al., 2019 [39]	GFCF experimental diet (6 months)	24 h recall Pre and post urinary concentrations of betacasomorphin	Regular diet	ECO scale	Null findings
Johnson et al., 2011 [36]	GFCF experimental diet (3 months)	Parents recorded food intake	Healthy diet without added sugar	CARS	Null findings
Knivsberg et al., 2002 [20]	GFCF experimental diet (12 months)	Pre and post urinary concentration of creatine	Regular diet		Improvement of core autistic symptoms on GFCF diet
Navarro et al., 2015 [38]	2 weeks of GFCF diet followed by 4 weeks of GFCF diet + supplement containing brown rice flour (4 weeks)	Daily food diary form completed by the parents.	2 weeks of GFCF diet followed by 4 weeks of GFCF diet + supplement containing 0.5 g/kg of gluten powder and 0.5 g/kg of non-fat dried milk	Core autistic symptoms, clinician-reported	Null findings
Whiteley et al., 2010 [37]	GFCF experimental diet (12 months)	Nutritionists monitored to ensure dietary compliance and nutritional intake (method not specified). Urinary excretion (details not reported)	Regular diet	ATEC	Clinically irrelevant improvement of core autistic symptoms on GFCF diet

Abbreviations: GFCF: Gluten- and casein-free; ECO: Ecological Communication Orientation; CARS: Childhood Autism Rating Scale; ADOS: The Autism Diagnostic Observation Schedule; ATEC: The Autism Treatment Evaluation Checklist Scale; ABC: Aberrant Behavior Checklist; ERC-III: Behavioral Summarized Evaluation (ERC-III) scale (Evaluation Resumé du Comportement, in French); CBCL: Child Behavior Checklist; DIPAB: Diagnosis of Psychotic Behavior in Children (Diagnose of Psykotisk Adfærd hos Børn, in Danish); GARS: Gilliam Autism Rating Scale; VABS: Vineland Adaptive Behavior Scale.

**Table 4 nutrients-13-00470-t004:** Summary of findings on GFCF versus usual diet.

Outcome (Timeframe)	Study Results and Measurements	Certainty of the Evidence (Justification for Ratings)
Clinician assessed core symptoms (Minimum 6 months)	SMD: −0.31 (CI 95% −0.89–0.27) Based on data from 120 patients in four studies Follow up: 3–12 months	Very low ^a,b,c,d^
Parent assessed functional level (Minimum 6 months)	MD: 0.61 (CI 95% −5.92–7.14) Based on data from 55 patients in one study Follow up: 12 months	Very low ^a,e^
Parent assessed conduct problems (Minimum 6 months)	MD: 1.80 (CI 95% −6.56–10.16) Based on data from 22 patients in one study Follow up: 3 months	Very low ^a,c,d^
Body mass index, kg/m^2^ (Minimum 6 months)	MD: 0.30 (CI 95% −1.81–2.41) Based on data from 63 patients in one study	Very low ^e,f^
Number of persons with gastrointestinal discomfort (Minimum 6 months)	RR: 2.33 (CI 95% 0.69–7.90) Based on data from 22 patients in one study Follow up: 3 months	Very low ^a,c,d^
Number of persons with side effects (Minimum 6 months)	RR: 1.89 (CI 95% 1.11–3.21) RD: 0.23 (CI 95% −0.26–0.72) Based on data from 94 patients in two studies Follow up: 3–12 months	Very low ^a,c,d^

CI: confidence interval; MD: mean difference; RD: risk difference; RR: risk ratio; SMD: standardized mean differences. Very low quality: we have very little confidence in the effect estimate. ^a^ serious risk of bias; ^b^ serious risk of inconsistency; ^c^ serious risk of indirectness; ^d^ serious risk of imprecision; ^e^ very serious risk of imprecision; ^f^ very serious risk of bias.

## Data Availability

Data is available at the Danish Health Authority website (www.sst.dk).

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
