# Peer review of "The Effect of a Combined Gluten- and Casein-Free Diet on Children and Adolescents with Autism Spectrum Disorders: A Systematic Review and Meta-Analysis"

_nutrients, 2021, doi:10.3390/nu13020470_

Round 1
Reviewer 1 Report
This systematic review and meta-analysis investigated the clinical effects of gluten-free and casein-free (GFCF) diets for the management of autism spectrum disorders (ASD) in children and adolescents. The authors claimed that there is no statistical significance between GFCF group and the control group in clinician-reported autism core symptoms, parent-reported functional level and behavioral difficulties of the child. In contrast, gastrointestinal adverse effects were found in the GFCF group. Furthermore, the authors stated that further high-quality RCTs are required in the future.
However, some concerns existed.
1. The authors pointed out that this systematic review was reported adhering to the Preferred Reporting Items for Systematic Review and Meta-analysis (PRISMA) checklist, thus, my judgment was based on the checklist items first. Also, a PRISMA checklist should be provided as a supplementary file in the manuscript.
- Abstract item 2: data source and systematic review registration number should be provided in the abstract.
- Method item 5: The web address of the actual protocol should be provided and the registration number. I try to find the actual protocol on the URL that the authors provided, however, I could not find the protocol on the website.
- Result item 17: the flow diagram of the selection process should be in the main text of the manuscript, not in the supplementary files. Please move it back to the main text.
- Result item 18: Study sample size should be in the characteristics table (sample size in the treatment group and control group).
- Discussion item 26: Implications for clinical practice and further research, and future directions should be discussed in depth.
2. Other comments
- The current management of ASD should be introduced in the introduction section. Has the GFCF diet been widely recommended by practitioners in clinical practice for children and adolescents with ASD? The authors should provide more information.
- It is important to note that only three databases were searched. Did the authors consider that the reason caused insufficient included studies maybe because of limited databases used? I suggested that more databases should be included, such as Web of Science, Science Direct, Scopus, Cochrane Central Register of Controlled Trials, ProQuest, AMED, and Google Scholar.
- The authors mentioned that ‘Children of 3-17 years of age diagnosed with ASD’ were included. However, in the result section, Description of the primary studies lines 173, ‘A total of 178 children aged between 2 and 18 years participated in the included studies’. Please checked and confirmed the ages of the included participants.
- References are required for the following sentences. Result Description of the primary studies lines 172 to 173, ‘Three RCTs were conducted in the United States of America (USA) and the others in Spain, 172 Norway, and Denmark, respectively.’; Page 12 Paragraph 1 lines 1 to 5, ‘The quality of the evidence for the primary outcome clinician-assessed core symptoms …due to insufficient time frame.’
Author Response
This systematic review and meta-analysis investigated the clinical effects of gluten-free and casein-free (GFCF) diets for the management of autism spectrum disorders (ASD) in children and adolescents. The authors claimed that there is no statistical significance between GFCF group and the control group in clinician-reported autism core symptoms, parent-reported functional level and behavioral difficulties of the child. In contrast, gastrointestinal adverse effects were found in the GFCF group. Furthermore, the authors stated that further high-quality RCTs are required in the future.
However, some concerns existed.
- The authors pointed out that this systematic review was reported adhering to the Preferred Reporting Items for Systematic Review and Meta-analysis (PRISMA) checklist, thus, my judgment was based on the checklist items first. Also, a PRISMA checklist should be provided as a supplementary file in the manuscript.
Thank you for your suggestion of including the PRISMA check list. We fully agree, and we have added this to supplemental material with a reference to this in the manuscript on page 4, line 90.
Abstract item 2: data source and systematic review registration number should be provided in the abstract.
We have now included the data source. The registration of the protocol on the Danish Health Authority website do not have a registration number, but instead we refer to the website.
On page 2 line 33-34 we have written the following:
The search was conducted up to January 2020 in Medline, Embase, Cinahl and Cochrane Library.
Method item 5: The web address of the actual protocol should be provided and the registration number. I try to find the actual protocol on the URL that the authors provided, however, I could not find the protocol on the website.
It should be available on the website at the Danish Health Authority www.sst.dk. However, we have also attached it to this reply to the reviewers (please refer to the end of this document).
Result item 17: the flow diagram of the selection process should be in the main text of the manuscript, not in the supplementary files. Please move it back to the main text.
Thank you for the suggestion. The three flow diagrams of systematic review, secondary studies and primary studies have now been moved back to the main text.
Result item 18: Study sample size should be in the characteristics table (sample size in the treatment group and control group).
We agree that this information is relevant for the readers and have therefore added this to the characteristics table.
Discussion item 26: Implications for clinical practice and further research, and future directions should be discussed in depth.
We agree that this is relevant to the readers, and the discussion has now been elaborated substantially, also concerning implications for clinical practice, further research and future directions.
We have now included a sentence from line 319 about clinical practice:
The finding of a risk of weight loss is important information for clinical practice, and the use of anthropometric growth measures (weight and height and calculation of body mass index) when evaluating the effect of the diet restriction should be emphasized.
And on line 354 further research and future directions are mentioned:
Clinical implications of the present findings may be careful consideration of introducing a GFCF diet to families with children with ASD, unless intolerance or allergy towards gluten and/or casein has been suspected or detected, based on standardized and validated measures.
Clinicians may inform families of the paucity of evidence of beneficial effects, and the increased incidence of gastrointestinal adverse effects should be taken into consideration, as well as the overall strain on the family of the added burden of introducing an elimination diet, which may be difficult to adhere to.
- Other comments
The current management of ASD should be introduced in the introduction section. Has the GFCF diet been widely recommended by practitioners in clinical practice for children and adolescents with ASD? The authors should provide more information.
We agree that this is relevant to the readers, and the introduction has now been elaborated substantially. The following have been added on line 52:
…..but in the last couple of decades, the role of the gastrointestinal system in the development of ASD has been a topic of interest based on the finding of high prevalence of gastrointestinal problems and disorders in individuals with ASD (Buie et al. 2010, Xu et al. 2018). Furthermore, children with ASD have been found to exhibit higher levels of proinflammatory cytokines following exposure to food proteins from gluten, casein, and soy, compared with controls (Jyonouchi et al. 2002), findings that have stimulated research investigating the link between food allergy and ASD (Jyonouchi et al. 2009, Li et al. 2021). Theories of mechanisms of action of the involvement of the immune system and the gastrointestinal system in the development of ASD are many and include the involvement of the gut-blood-brain barrier where by-products of the gut microbiota like lipopolysaccharides and short-chain fatty acids have been proposed to modulate the cytokine production (Fattorusso et al. 2019). Also, by-products from the microbiota have been suggested to influence the synthesis of neuropeptides like serotonin, and peptides from gluten and casein have been hypothesized to increase the activity of the opioid system [8]. These neuropeptides are thought to result in impairments in social behavior and communication, and thus may be involved in the pathogenesis of ASD.
Besides being hypothesized to be involved in the development of ASD, gluten- and casein-free diet started to be used by some families for a couple of decades ago based on case reports as a treatment for symptoms of ASD as well as for gastrointestinal symptoms. However, a Cochrane review from 2008 [10], based on two randomized controlled trials [11,12], concluded that the evidence for the efficacy of eliminating gluten and casein from the diet, purely based on autism as an indication, is uncertain. As research is still needed to elucidate the pathophysiologic mechanisms behind the relationship between ASD and gluten and casein diet, recommendations on diet restrictions is presently limited to individuals where allergic reactions or intolerance have been detected irrespective of a diagnosis of autism (Hyman, Levy & Myers 2020).
It is important to note that only three databases were searched. Did the authors consider that the reason caused insufficient included studies maybe because of limited databases used? I suggested that more databases should be included, such as Web of Science, Science Direct, Scopus, Cochrane Central Register of Controlled Trials, ProQuest, AMED, and Google Scholar.
As you can see from the search strategy, Cochrane was searched via Medline (line 14) and Embase (line 14) using “Cochrane.jw.”.
The search strategy follows the evidence based standard procedures of from the Danish Health Authority. Furthermore, the topic itself may not invite for more relevant databases to be included, mainly because the databases mentioned, are more or less overlapping with the databases searched. Also, prior to the development of the protocol for the national clinical guidelines at the Danish Health Authority, a broad search was performed after previous guideline covering interventions among children and youth with ASD. Moreover, content experts in the working group of the guideline were conferred, if any studies were missing. Finally references of existing SLR were also screened for relevant studies. Thus, although we can never exclude that we are missing smaller studies, we are confident in the search strategy, and including the above-mentioned databases would most likely not change the results.
This information has now been added to the manuscript in the abstract and line 100 under the subheading Search Strategy.
The authors mentioned that ‘Children of 3-17 years of age diagnosed with ASD’ were included. However, in the result section, Description of the primary studies lines 173, ‘A total of 178 children aged between 2 and 18 years participated in the included studies’. Please checked and confirmed the ages of the included participants.
Due to few numbers of primary studies, we decided to include the studies with mean ages below 3 years, which we have specified in the manuscript. Furthermore, the oldest participants were mean age 9 years, and we have therefore changed the upper limit of the age range.
On line 205 it now states:
A total of 178 children aged between 2 and 9 years participated in the included studies. Due to few numbers of primary studies, we decided to include studies with mean ages below 3 years.
References are required for the following sentences. Result Description of the primary studies lines 172 to 173, ‘Three RCTs were conducted in the United States of America (USA) and the others in Spain, 172 Norway, and Denmark, respectively.’; Page 12 Paragraph 1 lines 1 to 5, ‘The quality of the evidence for the primary outcome clinician-assessed core symptoms …due to insufficient time frame.’
References have now been provided.
Reviewer 2 Report
This is a well-written ambitious project. it is methodically organized in terms of approach and style. It appears to be an important objective contribution to the literature.
Author Response
This is a well-written ambitious project. it is methodically organized in terms of approach and style. It appears to be an important objective contribution to the literature.
Thank you
Reviewer 3 Report
The manuscript entitled “The effect of a combined gluten- and casein-free diet on children and adolescents with autism spectrum disorders: A systematic review and meta-analysis” presents interesting issue, but some areas must be corrected.
Major:
The main problem with the presented study is the fact that Authors declare that their systematic review and meta-analysis “adheres to the Preferred Reporting Items for Systematic Review and Meta-analysis (PRISMA)” (lines 66-67), but in fact the manuscript is not prepared according to the recommendations of PRISMA, which are very specific and should be rigorously followed. Authors should get familiar with PRISMA checklist (http://prisma-statement.org/prismastatement/Checklist.aspx) and they should correct their manuscript to be prepared according to the checklist. E.g., the Abstract should include: background, objectives, data sources, study eligibility criteria, participants, and interventions, study appraisal and synthesis methods, results, limitations, conclusions and implications of key findings, systematic review registration number, while a number of elements is not presented in the Abstract of the submitted manuscript. However, Authors should correct the whole study (not only the Abstract Section).
General:
Authors should prepare their manuscript according to the instructions for authors.
Abstract:
For the systematic review and meta-analysis readers this section is crucial and it requires major corrections to provide all the necessary information (see above).
Introduction:
Authors should properly justify conducting the systematic review for the gluten- and casein-free diet. The references which they present (sentence “the role of the intestinal system in the development of ASD has been a topic of interest, including the significance of gluten and casein [4–7]”) focus on the gut microbiota, so more accurate references are necessary.
Authors should avoid using the terms which are not scientific terms (“leaky gut”), but they should rather properly present the intestinal permeability phenomena.
Materials and Methods:
Authors should include flow chart of their searching procedure.
Authors should justify their studied group (why children and adolescents aged 3-17, not 3-18 were included?).
Authors should properly describe their methodology step by step with all necessary details.
Results:
The major part of the manuscript is one huge table (Table 1 – pages 6-8), which is very hard to follow and does not present the essential information. Authors should divide this huge table into several smaller ones and include to them also other important information form the included studies (e.g. country/ region, time when the study was conducted, tools which were used to monitor dietary intervention, detailed observations, conclusions formulated by authors of the included studies).
Figure 7 – as this figure reproduce the information presented above, it should be rather presented as a supplementary material.
Discussion:
Authors should broaden their discussion.
Authors should: (1) compare gathered data with the results by other authors, (2) formulate implications of the results of their study and studies by other authors, (3) formulate the future areas which should be studied.
Authors should deeply discuss the problem of the potential reverse causality. The main problem with the presented study is the risk of reverse causality, which was not taken into account properly. Authors assumed that gluten- and casein-free diet, as a result may influence the symptoms of autism spectrum disorders. However, it is also possible that the disease severity influences the possibility to follow any diet. This problem should be deeply discussed.
Author Response
The manuscript entitled “The effect of a combined gluten- and casein-free diet on children and adolescents with autism spectrum disorders: A systematic review and meta-analysis” presents interesting issue, but some areas must be corrected.
Major:
The main problem with the presented study is the fact that Authors declare that their systematic review and meta-analysis “adheres to the Preferred Reporting Items for Systematic Review and Meta-analysis (PRISMA)” (lines 66-67), but in fact the manuscript is not prepared according to the recommendations of PRISMA, which are very specific and should be rigorously followed. Authors should get familiar with PRISMA checklist (http://prisma-statement.org/prismastatement/Checklist.aspx) and they should correct their manuscript to be prepared according to the checklist. E.g., the Abstract should include: background, objectives, data sources, study eligibility criteria, participants, and interventions, study appraisal and synthesis methods, results, limitations, conclusions and implications of key findings, systematic review registration number, while a number of elements is not presented in the Abstract of the submitted manuscript. However, Authors should correct the whole study (not only the Abstract Section).
Thank you for your suggestion. We fully agree, and we have changed the manuscript accordingly and have added the PRISMA checklist to the supplemental material with a reference to this in the manuscript on page 4, line 67.
General:
Authors should prepare their manuscript according to the instructions for authors.
Thank you for your suggestion. We fully agree, and we have changed the manuscript accordingly.
Abstract:
For the systematic review and meta-analysis readers this section is crucial and it requires major corrections to provide all the necessary information (see above).
Thank you for your suggestion. We fully agree, and we have changed the abstract accordingly.
Introduction:
Authors should properly justify conducting the systematic review for the gluten- and casein-free diet. The references which they present (sentence “the role of the intestinal system in the development of ASD has been a topic of interest, including the significance of gluten and casein [4–7]”) focus on the gut microbiota, so more accurate references are necessary.
We agree that this is relevant to the readers, and the introduction has now been elaborated substantially. For more details see reply to Reviewer 1, in the section Other Comments.
Authors should avoid using the terms which are not scientific terms (“leaky gut”), but they should rather properly present the intestinal permeability phenomena.
Thank you for this comment. We agree, and the introduction has now been elaborated substantially and the term “leaky gut” is not mentioned anymore.
Materials and Methods:
Authors should include flow chart of their searching procedure.
Thank you for the suggestion. The three flow diagrams of systematic review, secondary studies and primary studies have now been moved back to the main text.
Authors should justify their studied group (why children and adolescents aged 3-17, not 3-18 were included?).
Thank you. This information is relevant to the readers and have been added on line 112 under specification of the population.
Authors should properly describe their methodology step by step with all necessary details.
Thank you for your suggestion. We fully agree, and we have changed the manuscripts accordingly
Results:
The major part of the manuscript is one huge table (Table 1 – pages 6-8), which is very hard to follow and does not present the essential information. Authors should divide this huge table into several smaller ones and include to them also other important information form the included studies (e.g. country/ region, time when the study was conducted, tools which were used to monitor dietary intervention, detailed observations, conclusions formulated by authors of the included studies).
Information on country/region and year are presented in the first column of the table “Study’s first author, year, country”.
Monitoring strategies of the dietary intervention of the studies are presented under “intervention”
Conclusions formulated by the authors are reported under “overall results”.
Figure 7 – as this figure reproduce the information presented above, it should be rather presented as a supplementary material.
Thank you, and we agree. Figure 7 has now been moved to supplementary material.
Discussion:
Authors should broaden their discussion.
Authors should: (1) compare gathered data with the results by other authors, (2) formulate implications of the results of their study and studies by other authors, (3) formulate the future areas which should be studied.
We agree that this is relevant to the readers, and the discussion has now been elaborated substantially, accordingly (from line 328):
The present review suggests that a GFCF diet may cause some increased incidence of gastrointestinal adverse effects which is important to bear in mind when choosing to put a child on a gluten- and/or casein-free diet which is often performed by parents in the hope of reducing or even eliminating autistic symptoms. Also, it is well-known that individuals with ASD commonly present selective eating patterns which may worsen when introducing a diet with subsequent risk of eating disorder and/or malnutrition [31–33].
Besides gastrointestinal side effects, it is worth noticing the risk of decreased appetite and weight loss in addition to waking up at night when introducing a gluten- and casein-free diet. Sleep disturbances are common in children with ASD (Hyman, Levy & Myers 2020), and the addition of diet restrictions to a child with ASD may worsen sleep difficulties and subsequently worsen the well-being of the child. The finding of a risk of weight loss is important information for clinical practice, and the use of anthropometric growth measures (weight and height and calculation of body mass index) when evaluating the effect of the diet restriction should be emphasized.
And line 355:
Clinical implications of the present findings may be careful consideration of introducing a GFCF diet to families with children with ASD, unless intolerance or allergy towards gluten and/or casein has been suspected or detected, based on standardized and validated measures.
Clinicians may inform families of the paucity of evidence of beneficial effects, and the increased incidence of gastrointestinal adverse effects should be taken into consideration, as well as the overall strain on the family of the added burden of introducing an elimination diet, which may be difficult to adhere to.
Authors should deeply discuss the problem of the potential reverse causality. The main problem with the presented study is the risk of reverse causality, which was not taken into account properly. Authors assumed that gluten- and casein-free diet, as a result may influence the symptoms of autism spectrum disorders. However, it is also possible that the disease severity influences the possibility to follow any diet. This problem should be deeply discussed.
We agree that this is relevant to the readers, and the discussion has now been elaborated substantially, accordingly (line 344):
The lack of efficacy of the diet may be explained by low adherence to the diet restriction due to severity of ASD in the child and/or parental difficulties with supplying GFCF food, possibly due to the added burden of a strict elimination diet in addition to the often stressed family life of having children with ASD (Bonis, 2016 REF). Five out of six included studies monitored adherence to diet (12,25, 26, 27 28), mainly relying on parental reports. In two included studies (12,28), where urinary concentrations of peptide concentrations were monitored in addition to parental monitoring of adherence of diet, there was no significant decrease in peptide concentrations following GFCF diet, suggesting either contamination no efficacy. This warrants caution regarding the conclusions that can be drawn from the null findings. It may be noteworthy, that if strict adherence is hard for children and families to follow, causing no effect of the elimination diet, this could point to a potential problem of feasibility of the diet itself.
Round 2
Reviewer 1 Report
The authors modified the manuscript according to the reviewer's instructions. However, two minor concers still existed.
- In terms of the last concern I provided last time, I am wondering whether the authors could clarify why they included the studies with mean ages below 3 years in the manuscript? Why not 2 years or 4 years?
- Why did the authors move Figure 7 to supplementary files? Suggest to keep Figure 7 (ROB) in the main text.
- References are required for this sentence in lines 70-72 'Besides being hypothesized to be involved in the development of ASD...as well as for gastrointestinal symptoms'.
- Line 383 'Research should carefully monitor participant adherence to the diet.' Researchers or Research? Please proofread the whole manuscript before resubmission.
Author Response
The authors modified the manuscript according to the reviewer's instructions. However, two minor concerns still existed.
In terms of the last concern I provided last time, I am wondering whether the authors could clarify why they included the studies with mean ages below 3 years in the manuscript? Why not 2 years or 4 years?
Thank you for the comment. We agree that the selected age may seem arbitrary, however since the guideline was performed in a Danish context, we selected the age range based on earliest age of diagnosis and the legal adult age in Denmark. This information has now been added to line 129:
This age range was chosen to cover children and adolescents with ASD from the earliest age of a valid diagnosis (3 years of age) until the 18th birthday where adolescents in Denmark come of age.
Why did the authors move Figure 7 to supplementary files? Suggest to keep Figure 7 (ROB) in the main text.
The move of Figure 7 was requested by reviewer 3. We have now moved figure 7 back to the main text.
References are required for this sentence in lines 70-72 'Besides being hypothesized to be involved in the development of ASD...as well as for gastrointestinal symptoms'.
Thank you. We have now provided references in line 70-72.
Line 383 'Research should carefully monitor participant adherence to the diet.' Researchers or Research? Please proofread the whole manuscript before resubmission.
Thank you for identifying this typo. The sentence has been changed to “researchers”.
The manuscript has now been proofread.
Reviewer 3 Report
The manuscript entitled “The effect of a combined gluten- and casein-free diet on children and adolescents with autism spectrum disorders: A systematic review and meta-analysis” presents interesting issue, but some areas must be corrected. Unfortunately, Authors sis not include required comments, even if they stated that they did so.
Major:
The main problem with the presented study is the fact that Authors declare that their systematic review and meta-analysis “adheres to the Preferred Reporting Items for Systematic Review and Meta-analysis (PRISMA)” (lines 66-67), but in fact the manuscript is not prepared according to the recommendations of PRISMA, which are very specific and should be rigorously followed. Authors did not correct it properly. Authors should get familiar with PRISMA checklist (http://prisma-statement.org/prismastatement/Checklist.aspx) and they should correct their manuscript to be prepared according to the checklist. E.g., the Abstract should include: background, objectives, data sources, study eligibility criteria, participants, and interventions, study appraisal and synthesis methods, results, limitations, conclusions and implications of key findings, systematic review registration number, while a number of elements is not presented in the Abstract of the submitted manuscript. However, Authors should correct the whole study (not only the Abstract Section).
General:
Authors should prepare their manuscript according to the instructions for authors. Authors did not correct it properly.
Abstract:
For the systematic review and meta-analysis readers this section is crucial and it requires major corrections to provide all the necessary information (see above). Authors did not correct it properly.
Materials and Methods:
Authors should justify their studied group based on the proper literature (why children and adolescents aged 3-17, not 3-18 were included?).
Results:
The major part of the manuscript is one huge table (Table 1 – pages 6-8), which is very hard to follow and does not present the essential information. Authors should divide this huge table into several smaller ones and include to them also other important information form the included studies (e.g. country/ region, time when the study was conducted, tools which were used to monitor dietary intervention, detailed observations, conclusions formulated by authors of the included studies).
Author Response
The manuscript entitled “The effect of a combined gluten- and casein-free diet on children and adolescents with autism spectrum disorders: A systematic review and meta-analysis” presents interesting issue, but some areas must be corrected. Unfortunately, Authors sis not include required comments, even if they stated that they did so.
Major:
The main problem with the presented study is the fact that Authors declare that their systematic review and meta-analysis “adheres to the Preferred Reporting Items for Systematic Review and Meta-analysis (PRISMA)” (lines 66-67), but in fact the manuscript is not prepared according to the recommendations of PRISMA, which are very specific and should be rigorously followed. Authors did not correct it properly. Authors should get familiar with PRISMA checklist (http://prisma-statement.org/prismastatement/Checklist.aspx) and they should correct their manuscript to be prepared according to the checklist. E.g., the Abstract should include: background, objectives, data sources, study eligibility criteria, participants, and interventions, study appraisal and synthesis methods, results, limitations, conclusions and implications of key findings, systematic review registration number, while a number of elements is not presented in the Abstract of the submitted manuscript. However, Authors should correct the whole study (not only the Abstract Section).
The abstract and the remaining text has now been critically revised according to the recommendations of PRISMA.
As an example, the abstract has been changed to the following:
There has been a growing interest in the gastrointestinal system and its significance for autism spectrum disorder (ASD), including the significance of adopting a gluten-free and casein-free (GFCF) diet. The objective was to investigate beneficial and safety of a GFCF diet among children with a diagnosis of ASD. We performed a systematic literature search in Medline, Embase, Cinahl and Cochrane Library up to January 2020 for existing systematic reviews and individual randomized controlled trials. Studies were included if they investigated a GFCF diet compared to a regular diet in children aged 3 to 17 years diagnosed with ASD, with or without comorbidities. The quality of the identified existing reviews was assessed using the AMSTAR tool. The risk of bias in RCTs was assessed using the Cochrane Risk of Bias Tool and overall quality of evidence was evaluated using GRADE. We identified six relevant RCTs including 143 participants. The results from a random effect model showed no effect of a GFCF diet on clinician-reported autism core symptoms (SMD -0.31 [95% Cl. -0.89, 0.27]), parent-reported functional level (MD 0.61 [95% Cl. -5.92, 7.14 ]) or behavioral difficulties (MD 0.80 [95% Cl. -6.56, 10.16 ]). On the contrary, a GFCF diet might trigger gastrointestinal adverse effects (RR 2.33 [95% Cl. 0.69, 7.90]). The quality of evidence ranged from low to very low, due to serious risk of bias, serious risk of inconsistency, and serious risk of imprecision.
Clinical implications of the present findings may be careful consideration of introducing a GFCF diet to children with ASD. However, the limitations of the current literature hinder the possibility of drawing any solid conclusion and more high-quality RCTs are needed. The protocol is registered at the Danish Health Authority website (www.sst.dk).
General:
Authors should prepare their manuscript according to the instructions for authors. Authors did not correct it properly.
We have now changed the reference list, and the authors contribution to the manuscript, as well as provided details on the informed consent statement and data availability. Moreover, the figure format now follows the instructions for authors,
Abstract:
For the systematic review and meta-analysis readers this section is crucial and it requires major corrections to provide all the necessary information (see above). Authors did not correct it properly.
Please refer to the reply above.
Materials and Methods:
Authors should justify their studied group based on the proper literature (why children and adolescents aged 3-17, not 3-18 were included?).
Thank you for the comment. We agree that the selected age may seem arbitrary, however since the guideline was performed in a Danish context, we selected the age range based on earliest age of diagnosis and the legal adult age in Denmark. This information has now been added to line 129:
This age range was chosen to cover children and adolescents with ASD from the earliest age of a valid diagnosis (3 years of age) until the 18th birthday where adolescents in Denmark come of age.
Results:
The major part of the manuscript is one huge table (Table 1 – pages 6-8), which is very hard to follow and does not present the essential information. Authors should divide this huge table into several smaller ones and include to them also other important information form the included studies (e.g. country/ region, time when the study was conducted, tools which were used to monitor dietary intervention, detailed observations, conclusions formulated by authors of the included studies).
Thank you for the suggestion. The table has now been split up in three tables: Table 1 regarding study identification. Compared to the previous table, there is now added new information on region or city, and trial registration date (as a proxy of time when the study was conducted) otherwise we refer to publication date. Table 2 regarding baseline characteristics. Compared to the previous table we have added information on exclusion criteria. Table 3 regarding the intervention, control, reported outcomes and authors conclusion. Compared to the previous table there is new information on monitoring dietary intervention.